# Single-Cell Transcriptional Landscape Reveals the Regulatory Network and Its Heterogeneity of Renal Mitochondrial Damages in Diabetic Kidney Disease

**DOI:** 10.3390/ijms241713502

**Published:** 2023-08-31

**Authors:** Chenhua Wu, Yuhui Song, Yihong Yu, Qing Xu, Xu Cui, Yurong Wang, Jie Wu, Harvest F. Gu

**Affiliations:** 1Laboratory of Molecular Medicine, School of Basic Medicine and Clinical Pharmacy, China Pharmaceutical University, Nanjing 210009, China; 3121030170@stu.cpu.edu.cn (C.W.); 3221091994@stu.cpu.edu.cn (Y.S.); 3222031034@stu.cpu.edu.cn (Y.Y.); 3320092077@stu.cpu.edu.cn (Q.X.); 3321091932@stu.cpu.edu.cn (X.C.); 1620154332@cpu.edu.cn (Y.W.); 2Laboratory of Minigene Pharmacy, School of Life Science and Technology, China Pharmaceutical University, Nanjing 211198, China

**Keywords:** bioinformatics, diabetic kidney disease, mitochondria, single-cell RNA sequencing, transcriptional regulation

## Abstract

Diabetic kidney disease (DKD) is one of the common chronic microvascular complications of diabetes in which mitochondrial disorder plays an important role in its pathogenesis. The current study delved into the single-cell level transcriptome heterogeneity of mitochondrial homeostasis in db/db mice, an animal model for study of type 2 diabetes and DKD, with single-cell RNA sequencing (scRNA-Seq) and bulk RNA-seq analyses. From the comprehensive dataset comprising 13 meticulously captured and authenticated renal cell types, an unsupervised cluster analysis of mitochondria-related genes within the descending loop of Henle, collecting duct principal cell, endothelial, B cells and macrophage, showed that they had two types of cell subsets, i.e., health-dominant and DKD-dominant clusters. Pseudotime analysis, cell communication and transcription factors forecast resulted in identification of the hub differentially expressed genes between these two clusters and unveiled that the hierarchical regulatory network of receptor-TF-target genes was triggered by mitochondrial degeneration. Furthermore, the collecting duct principal cells were found to be regulated by the decline of *Fzd7*, which contributed to the impaired cellular proliferation and development, apoptosis and inactive cell cycle, as well as diminished capacity for material transport. Thereby, both scRNA-Seq and bulk RNA-Seq data from the current study elucidate the heterogeneity of mitochondrial disorders among distinct cell types, particularly in the collecting duct principal cells and B cells during the DKD progression and drug administration, which provide novel insights for better understanding the pathogenesis of DKD.

## 1. Introduction

The latest figures released by the International Diabetes Federation in 2021 show that 537 million (1 in 10) adults now live with diabetes worldwide; a rise of 16% (74 million) since the previous estimates in 2019 [1]. Due to the microangiopathy caused by hyperglycemia, up to 30~50% of diabetic patients have diabetic kidney disease (DKD) [2]. DKD has been recognized as the uppermost cause of end stage kidney disease, which brings tremendous health and property deficiency to patients and medical finance.

Mitochondria exhibit a semi-autonomous nature due to their double-membrane structure. They possess their own genome and function as the cellular energy-generating factories, providing over 90% of the cell’s ATP supply. Mitochondria can oxidize small molecular organic acids, fatty acids and ketones to produce energy, extensively participating in the regulation of cell metabolism, energy supply, cell cycle and other intracellular activities. Numerous studies have demonstrated that the mitochondrial disorder caused by metabolic changes of diabetes is one of the important causes of DKD [3]. In DKD, mitochondrial dysfunction has a definite relation with DKD because of the reduced quantity, impaired oxidative function, kinetic imbalance and fuel transition [4]. Furthermore, the oxygen contents vary across different regions of the kidney, leading to diverse metabolic characteristics of mitochondria in various types of renal cells. Consequently, the changes of mitochondria in distinct cell types at different stages of DKD may exhibit variations as well. Therefore, it is imperative to carry out a comprehensive analysis of mitochondria considering heterogeneity among different cell types in kidney of DKD at single-cell resolution.

Bulk RNA-Seq, known for its high-throughput capabilities, has been a powerful technology for screening biomarkers or therapeutic targets for diverse tissues and organs [5]. Zhao et al. utilized this technology to investigate the biomarkers of DKD in db/db mice and found the mitochondrial dysfunction, abnormal lipid metabolism and oxidative stress [6]. Bulk RNA-Seq, however, has a limitation owning to the internal heterogeneity within tissues and organs. In recent years, single-cell RNA sequencing (scRNA-seq) has been widely used in the research of diseases such as hepatology [7], cardiovascular disease [8], and neurodegenerative disease [9]. Compared to bulk RNA-Seq, scRNA-Seq advantageously offers the ability to comprehensively consider the analysis of heterogeneity among the types of cells. Leveraging scRNA-Seq, we recently identified the cell-specific targets involved in the aetiology of DKD in db/db mice [10].

In the current study, we performed an extensive analysis of the mitochondrial landscape in DKD based upon the data of specific cell types of scRNA renal transcription to reveal the regulatory network and its heterogeneity of renal mitochondrial damages in distinct renal cell types of DKD. There are universal transitions of material transportation, metabolism, cell cycle in the descending loop of Henle (DLH), collecting duct principal cells (CD-PC), endothelial cells (EnCs), B cells and macrophages (Mac). Especially notable is the fluctuation in the CD-PC and B cells, whose cell state can be reversed by angiotensin receptor blockers (ARB) and Huangkui capsule (HKC).

## 2. Results

### 2.1. Pathological Features and Reactive Oxygen Species

According to the HE staining, compared with Ctrl mice, the DKD group showed thickening and diffuse hyperplasia of glomerular basement membrane, early Kimmelstiel–Wilson (KW) nodule formation, glomerular atrophy, transparent capillary lesions and other characteristics (Figure 1A). Meanwhile, necrosis and exfoliation of renal tubular epithelial cells, vacuolar lesions and dilatation of renal tubular lumen were observed in Periodic Acid-Schiff (PAS)-staining of the DKD group (Figure 1B). The images showed the typical pathological manifestations in kidney of DKD. Notably, the increase of fluorescence intensity of reactive oxygen species (ROS) fluorescence in DKD was obvious (Figure 1C). Based upon the electron microscope observation, we further found that most mitochondria in the Ctrl group were abundant in number, long and complete in shape, with dense mitochondria matrix and intima formed crista, but mitochondria in the DKD group were mostly round, with transparent matrix, loose in structure, intima and crista (Figure 1D).

### 2.2. Renal scRNA-Seq Data Preparation, Clustering, and Quality Control

These cells were divided into 13 distinct cell types, including ascending loop of Henle (ALH), Collecting duct intercalated cell (CD-IC), CD-PC, Distal convoluted tubule (DCT), DLH, EnC, Mac and segment 1 (S1), S2, S3 of PTCs (Figure 2A,B). The top 10 markers of each cell type showed no clustering bias caused by batch effect between groups (Figure 2C,D). Mitochondrial RNA content is typically associated with cell survival rates during the preparation of a single-cell suspension and the quality of scRNA-Seq library construction. To mitigate any potential interference arising from cell survival rates and RNA library integrity, the double droplet judgment was carried out.

### 2.3. Mitochondrial Encoding mRNA Content

As a semi-autonomous organelle with its own diminutive genome, mitochondria contains 37 self-encoding genes, of which 13 encode mRNA. Current scRNA-Seq expression matrix included 13 mRNA genes encoded by mitochondria, and the expression of these 13 mRNA genes in PTC and other cells was summarized (Appendix A).

To explore whether the expression pattern encoded by mitochondria was involved in the progress of DKD, we counted the expression of these 13 genes in each cell type (Figure 2E,F, Appendix A) and found that the percentages of mitochondrial encoding mRNA (PMEM) in PTCs (S1: 7.89%; S2: 9.51%; S3: 7.29%) and collecting tubules (CD-IC: 10.7%; CD-PC: 7.85%), which were higher than other cells. This might reflect the function of genes in these PTC, CD-IC, and CD-PC cells to transport large amounts of organic and inorganic ions between blood and initial urine. The overall distribution of gene set variation analysis (GSVA) scores between the Ctrl and DKD groups at the whole kidney level (Figure 3A) showed insignificant differences. This phenomenon might be caused by ignoring the heterogeneity among renal cell types. Therefore, we further conducted an analysis of inter-group genetic difference at the levels of whole kidney and PTC. The intersection of the two differential analyses showed that the differential expressed genes in the whole kidney (Wilcox test, *p* < 0.05|log_2_(FC)| > 0.5) were nearly identical to those in PTC (Figure 3B). It suggested that previous studies by using bulk RNA-seq in kidney had focused on PTC specifically. Obviously, the comparison of expression at the whole kidney level holds limited significance. We thus performed the analysis of difference between the Ctrl and DKD groups at the cell type level. In the S1 segment of PTCs, there was no significant difference between the groups. In the S2 segment of PTCs, the GSVA score of the mitochondrial coding gene set in the DKD group was significantly higher compared to the Ctrl group, while the GSVA score in the S3 segment of PTCs in the DKD group was significantly lower (Figure 3C). These findings were consistent with the trends in other cell types, except for CD-IC and DC, suggesting that PTCs were less affected by DKD than distal renal stromal cells (DRSC) and immune cells (ICs). Therefore, we re-counted the GSVA score of these cells at levels of PTCs, DRSC and ICs. In the different classifications, compared to the Ctrl group, GSVA score in PTCs of the DKD group was higher, while the GSVA score in DSRC and ICs were lower, and the downscaling multiples in ICs were larger and more significant (Figure 3D).

### 2.4. Mitochondria-Associated Nuclear Genes Were Involved in the Process of DKD

Widespreadly, more than one thousand mitochondria-associated nuclear genes (MANGs) are known to encode the mitochondrial components. Even the oxidative respiratory electron transport chain is located on the inner membrane of mitochondria and is heavily encoded by the MANGs (Appendix A). A total of 1140 MANGs were adopted from Mouse Mito Carta [11] and categorized into 151 pathways, such as mitochondrial central dogma, mtDNA maintenance, mtDNA replication, etc. (Appendix A). We used the pathways to guide the GSVA and subsequent difference analysis (generalized linear models) of each cell type (Appendix A). The GSVA score of 151 pathways showed the extensive differences as the GSVA scores of MANGs among different cell types between Ctrl and DKD groups. 

We assumed that there were healthy and DKD individuals in renal cells of DKD. Therefore, we used the GSVA score of significantly different pathways to re-cluster cell population in Seurat and divided them into two clusters (Appendix A) based upon the parameters (Table 1). We weighted the proportion and found the special clusters of S1, S3, DLH, ALH, CD-PC, CD-IC, EnC, B cells and Mac were significantly higher in the DKD group (Figure 4A). These subpopulations, called DKD-dominant Clusters (DDCs), were significantly higher than those in the Ctrl group. Tentatively, we referred to remaining subpopulations as health-dominant clusters (HDCs). We then conducted two difference analyses with the Wilcox test either between HDC cells of Ctrl group and DDC cells of the DKD group (different analysis 1, DA1) or between HDC cells and DDC cells in the DKD group (different analysis 2, DA2) of S1, S3, DLH ALH, CD-PC, CD-IC, EnC, B cells and Mac. After excluding MANGs, significantly differential genes were defined (*p*-value < 0.05 and |Log2(FC)| > 0.5). DLH, CD-PC, EnC, B cells and Mac shared the different genes, and the exclusive different genes in DA1 and DA2 accounted for more than 100 individuals (Figure 4B). The cell types with insufficient intersection genes were discarded. The Log2(FC) of the intersecting genes was presented in a scatter fitting curve to reflect their linear correlation (Figure 4C). There was a significant positive correlation within five cell types between the intersection genes of two comparisons. It was preliminarily speculated that the development of DLH, CD-PC, EnC, B cells and Mac in the formation of DKD might be driven by MANGs. Therefore, we extracted the intersection genes of two comparisons and exclusive gene of DA2 in DLH, CD-PC, EnC, B cells and Mac to perform a gene enrichment analysis based upon 51 DKD-conventional pathways (DCP) as we previously described [10]. The enrichment analysis showed that the intersection genes were DCP-enriched in DLH, CD-PC, EnC, B cells and Mac (Figure 4F, Appendix A). We selected the most significant 20 pathways (if sufficient) in DCP enrichment data of intersection genes and exclusive genes of DA2. The number of top 20 enriched DCP was decreased in the exclusive genes of DA2 in DLH and CD-PC even though we followed the same process and parameters in enrichment analysis (Figure 4F). To avoid that, HDC and DDC cells were introduced due to the errors in upstream cell annotation, the Person correlation analysis was performed among ten subsets of five cell types in DKD and corresponding to the cell types in Ctrl by using normalized expression matrix (Figure 4E). We found that all ten subclusters had the similarity with original five cell types. The subclusters might have the characteristics of DKD but the rest of the cell subclusters (HDCs) had the characteristics of normal samples. Furthermore, the exclusive genes of DA2 provided us with information other than conventional DCP. We performed a cross-analysis of ESDEGs in these five cell types (Figure 4D). The fold change and significance (*p* value, Wilcox test) between HDCs and DDCs in the respective cell types displayed intense heterogeneity (Figure 5A).

To explore the biological significance of ESDEGs, we performed a semi-supervised pseudotime analysis of these five cell types (Figure 5B–D). The trajectory inference preliminarily predicted that there was imbalance in the distribution of HDCs and DDCs, and the density curves of these cells in the pseudotime were significantly different (*p*-value < 0.001). To avoid the prior influence caused by the intervention of mitochondria-related pathways, MANGs were eliminated from the DA1 and DA2 when the ESDEGs were defined. To demonstrate that the pseudotime state differences between HDCs and DDCs are indeed associated with mitochondria, we summarized DKD-related mitochondrial pathways and corresponding genes previously confirmed to be associated with DKD for ssGSEA, a supervised machine learning approach [12]. The significantly different pathways were very consistent (Figure 5E). The ssGSEA score of mitochondrial and DKD-related pathways in DLH and CD-PC, which are responsible for the transport of renal tubular substances, showed a trend of significant up-regulation in most DDCs. However, these pathways in EnC, B cells and Mac, which are not responsible for the transport of renal tubular materials, mostly showed a downregulation trend in DDCs. It can be speculated that HDCs and DDCs have indeed changed in mitochondrial function and change of cell biological status driven by mitochondria, so the biological significance of ESDEGs is worth further revealing.

### 2.5. SLC Superfamily Intervened in the Process of DKD by Affecting Cellular Metabolism and Oxidative Stress through Substance Transport Function

There are a total of 364 members of solute carrier (SLC) superfamilies in mice. These SLC genes possess conserved domains and are widely distributed in various chromosomes (Appendix A), tissues and organs of the body, including kidney, liver, intestinal tract, etc. Functionally, the substances that SLCs transfer include the signaling molecules (such as cAMP, prostaglandins, bile acids and short-chain fatty acids), metabolites (such as *α*-ketoglutarate), and antioxidants (such as urate, ergothioenine, vitamins and cofactors), etc. [13]. We summarized the highly expressed SLCs in renal cells (Appendix A). The members, which were highly expressed in specific cells and acted as material carriers for the transport function of kidney cells are listed in Table 2.

The members of SLC25 family, as mitochondrial carriers, occur only in mitochondria [14]. Therefore, we identified the differently expressed genes (DEGs) of SLC25 members between HDC and DDC of the five cell types and retrieved their respective substrates (Table 3 and Appendix A). *Slc25a4* (ATP/ADP transport), *Slc25a5* (adenine nucleotide transport), *Slc25a30* (C4-dicarboxylate and sulfur compound transport) and *Slc25a3* (phosphate transport) as uncoupling proteins were found to be increased in the DDCs of CD-PC. Meanwhile, *Slc25a39* responsible for glutathione transport showed the same up-regulation trend in the DDCs of CD-PC. The remaining SLC25 DEGs’ transport substances were involved in oxidative phosphorylation such as nicotinamide adenine dinucleotide, ATP, nucleotide, malate, oxaloacetate and succinate, which participate in the tricarboxylic acid cycle. In addition, *Slc25a15*, the carrier of L-arginaine, L-Lysine and L-ornithine, showed an obvious down-regulation trend in the DDCs of EnC and B cells. The substances transported by the SLC25 family showed significant differences between DDCs and HDCs in oxidative phosphorylation, tricarboxylic acid cycle and amino acid metabolism in the mitochondria of CD-PC, EnC, B cells and DLH.

In addition to the members of the SLC25 family, there were 70 SLC superfamily members belonging to DEGs between HDCs and DDCs (Table 4 and Appendix A). This expands the differences in material transport. *Slc27a2* (long-chain fatty acid), *Slc5a8* (short-chain fatty acids), *Slc10a2* (bile acid-sodium), *Slc51b* (bile acid) and *Slco1a6* (bile acid) involved in lipid transport can be observed as DEGs. In terms of carbohydrate transporters, *Slc5a10* (glucose), *Slc5a2* (glucose), *Slc2a5* (fructose, glucose), *Slc2a2* (glucose), *Slc37a4* (glucose-6-phosphate), *Slc2a4* (glucose), *Slc2a1* (glucose), *Slc35a1* (pyrimidine nucleotide-sugar) and *Slc35a4* (pyrimidine nucleotide-sugar) were observed in DEGs of the corresponding cell types. Amino acids and their derivatives transporters also contained a large amount of DEGs, such as *Slc6a18*, *Slc7a13* (L-cystine, L-glutamate, aspartate), *Slc3a1* (neutral and basic amino acids), *Slc7a12*, *Slc6a13* (taurine), *Slc38a2* (L-glutamine, L-serine), *Slc38a3* (L-glutamine), *Slc6a19* (neutral amino acids), *Slc6a6* (*β*-alanine, taurine) and *Slc1a1* (aspartate, glutamate, chloride, cysteine). These findings suggested that the SLC superfamily contributed significantly to the heterogeneity of DDCs and HDCs through the transport of metabolites, which may affect the mitochondrial homeostasis in the progress of DKD.

### 2.6. Differential Expression of ESDEGs Was Controlled by a Hierarchical Receptor-TF-TG Regulatory Network Formed in Cell Communication

To further discover the biological insight of ESDEGs, the enrichment analyses of Biological Process, Molecular Function and Cell Component in Gene Ontology (GO) for these exclusive ESDEGs were performed and the 10 most significant pathways (if sufficient) for each cell type are shown (Figure 6A). Among them, the ESDEGs of DLH were enriched in the pathways, including wound healing, cell-substrate adhesion and regulation of actin filament-based process. For CD-PC, in addition to the enrichment of wounding-related pathways, epithelial proliferation-related pathways, and negative regulation of cell projection organization, the presence of negative regulation of phosphorylation might indicate that mitochondrial function had been disrupted. The same is true for the enrichment pathways of EnC, with the difference mainly reflected in the emergence of pathways related to the metabolism of amino acids, organic acids, carbohydrates, purines and other substances. In addition, these two pathways related to antigen presentation, namely antigen processing and presentation of peptide antigen via MHC class II and antigen processing and presentation of peptide or polysaccharide antigen via MHC class II, were enriched in B cells and pathways related to organic hydroxy and vitamin metabolism also appeared. Similar to B cells, the antigen processing and presentation of peptide antigen via the MHC Class I pathway was enriched in Mac, and the amino acid, sulfur compound, organic hydroxy and other metabolic pathways are still enriched in Mac. The GO pathways enriched by ESDEGs in these cells were similar to each other in the metabolism pathways, but each pathway has its own unique function.

According to the results of cell communication prediction, 46 pathways and 90 ligand-receptor interactions were summarized in Figure 6B,D and Appendix A. We found that DDCs and HDCs of B cells might play different roles in the incoming interaction strength. Besides, the DDCs and HDCs of DLH, EnC and CD-PC showed differences in incoming and outgoing to a certain extent (Figure 6C). Subsequently, the different cell communication pathways (DCCPs) contained the corresponding ESDEGs as the receptor of the incoming pathway or the ligand of the outgoing pathway in these five cell types (Table 5 and Figure 6E). Among the 10 subclusters, only the HDCs of EnC, DDCs of EnC and DDCs of Mac did not contain DCCP when acting as the target, while DLH contains DCCP whether in HDCs or DDCs (Appendix A). The DCCP in various cell species included APP, COLLAGEN, AGRN, JAM, LAMININ, etc. Most of them were found to be related with DKD or immunity [10], except AGRN communication pathway. For example, the COLLAGEN pathway is widely believed to be related to collagen deposition in kidney, and this pathway may lead to the occurrence of renal fibrosis in chronic kidney disease (CKD) and DKD [15]. TGF-*β*, as a well-known cytokine related to immune regulation, is widely expressed in renal tissues and involved in a variety of DKD characteristics such as extracellular matrix deposition, renal fibrosis, glomerular basement membrane expansion, and TGF-*β* pathway appeared as DCP when the DDCs of EnC and B cells acted as the source [15]. Among these, DCCP, ANGPTL (target is DLH), ncWNT (target is CD-PC) and JAM (targets are CD-PC and Mac) receptor genes exist as ESDEGs in four corresponding target cell types (Figure 7A and Table 5).

We further constructed the inferred gene regulatory network by SCENIC [16]. There were many effective regulons and differences between the DDCs and HDCs of five cell types (Appendix A). Among them, *Sdc4* of DLH drove 8 downstream transcription factors (TF), resulting in fluctuations in the expression of 167 downstream target genes (TG). As the receptors, *Fzd7* and *F11r* regulated 115 TG by 29 TF in CD-PC while *F11r* regulated 8 TF to cause a fluctuation in 170 TG of Mac. As an important marker of B cells, *Cd74* acted as a receptor to regulate 31 TF and affected the expression of 286 downstream TF (Figure 7B). The complete regulatory networks and correlation of TF and TG downstream regulated by ESDEG receptor of these four cell types suggested that fluctuations of ESDEG were actuated by DCCP in corresponding cell (Figure 5A and Appendix A).

To understand the biological processes are controlled by these ESDEG receptors through TF, we performed Kyoto Encyclopedia of Genes and Genomes (KEGG) analysis on the regulon containing more than 50 TG (Figure 8A and Appendix A). DLH demonstrated the regulation of actin and cell adhesion gene sets resulting from the regulation of *Sdc4*. CD-PC contained abundant TF downstream of *F11r* and *Fzd7*. TF regulated by *Fzd7* mainly mediate epidermis development, epidermal cell differentiation and establish or maintain the cell polarity, extrinsic apoptotic signaling pathway, epidermal cell differentiation and other pathways. This implicated that the DDCs of CD-PC might be subject to the regulation of cell fate. *Fzd7* is highly expressed in renal mesenchymal during renal development and is one of the markers of renal stem cells [17]. Repressor of *Fzd7* may lead to cell apoptosis and death [18]. Meanwhile, the cell cycle prediction by using a cell cycle signature demonstrated that the proportion of G2M phase in DDCs of CD-PC during the active cell cycle was significantly lower than that of HDCs (Figure 8B), and the G2M score showed the same trend (Figure 8C). Intriguingly, the cibersort score of DDCs in CD-PC was significantly lower than that of DKD group in the same hyperglycemic diabetes mellitus (DM) group, whose urinary albumin-to-creatinine ratio (UACR) did not meet the DKD standard, while the HDCs of CD-PC showed the opposite trend (Figure 8E–G). After treatment with HKC, ARB, and SGLT2i, the cell ratio in the scRNA-Seq data showed that HKC and ARB could prevent the increase of DDC ratio in CD-PC, while SGLT2i could not (Figure 8D). In B cells, the KEGG pathways regulated by *Cd74* were mainly involved in ribosomes, endoplasmic reticulum and related to protein synthesis. As a pro-inflammatory factor, the activation of *Cd74* in B cells implies that it is induced to enter the S phase of cell cycle and improves the ability to synthesize DNA [19]. In the current study, we observed a corresponding phenomenon in the cell cycle prediction. The DDCs of B cells contained more S-phase cells (Appendix A) and obtained a higher S phase score (Appendix A) than that of HDCs. Besides, whether it is SGLT2i, ARB, or HKC, the HDC level of the B cells can be significantly recovered (Appendix A). For Mac, the downstream genes mainly exhibited the significant enrichment pathways regulated by *F11r* in the binding of Cd8, *β*-2-microglobulin, TAP1, TAP2 and other molecules and protein metabolism (Appendix A). Taken altogether, the data suggested that the HDCs and DDCs were distinctly regulated by DCCP and ligand-receptors.

## 3. Discussion

In the current study, we have uncovered that the mitochondrial contents in the kidney were not consistent within the cell types. Instead, the mitochondrial contents showed the distinct trends in different parts of the nephron and infiltrative IC, and the fold changes of mitochondrial content were significantly decreased in DSRC and IC of DKD than what in PTCs (Figure 3D). Subsequently, DRSC demonstrated the most differences in mitochondrial transcriptome. Therefore, our study has provided evidence that the different cell types of kidney may have the different degrees of mitochondrial stress.

The heterogeneity among various renal cell types in DKD may not only be inherent but also pathological. To verify the occurrence of normal-like subpopulation among the renal cell of DKD, we have assuredly identified two cell subsets (HDC and DDC) in DLH, CD-PC, EnC, B cells and Mac by scoring 151 mitochondria-related pathways. The differential genes between DDCs and HDCs overlap with the differences between DKD and Ctrl groups to a certain extent, and these overlaps have an intensively positive correlation (Figure 4C). Furthermore, the enrichment results showed that the overlap genes can be divided into DCP (Figure 4F). Although MANGs were excluded when the intersective genes were enriched to DCP, we have demonstrated that several pathways, especially oxidation and metabolism pathways, may be indirectly related to mitochondrial function in current research. For instance, both ROS elimination and ROS GSEA pathways were observed in the five cell types, suggesting that the excessive production of ROS in DKD would lead to oxidative stress in the microenvironment to stimulate the expression of TGF-*β* [15], and further cause the mitochondrial morphological deterioration, apoptosis, extracellular matrix degradation, mesangial dilation, glomerular basement membrane thickening, renal tubule fibrosis and other characteristics (Figure 1). Indeed, we have detected the emergence of apoptosis, TGF-*β*, NF*κ*B, epithelial-mesenchymal transition (EMT) and other related pathways (Figure 4F). In addition, the metabolic pathways related to amino acids, glucose, fatty acids and their derivatives were also enriched. As we have previously reported, there are indeed metabolite differences in the peripheral blood of db/db mice with DKD [20]. These disorders in carbohydrates, amino acids and fats could be further observed in the kidneys, which is not just limited in the peripheral blood. Carbohydrates and lipids, as the principal energy sources, may have varying availabilities in different types and phases of cells. In healthy renal tubulointerstitium, ATP is produced by the oxidation of free fatty acids and ketones, except for glomerulus [3]. Under the induction of diabetic high glucose environment; however, the damaged renal tubules may undergo metabolic transformation from oxidative phosphorylation of lipid metabolism to glycolysis, which will further lead to lipid accumulation. This will cause further oxidative impairment in renal stromal cells, and the accumulation of lipid in Mac will inhibit the autophagy of macrophages, thus inhibiting their ability to transform from M1 (immune activation) to M2 (immunosuppression), and ultimately leading to the increase of local inflammation activity [21]. Based upon the ssGSEA score of mitochondria-related DKD pathways, we have demonstrated that the glucose and lipid metabolism-related pathways are regulated differently across distinct cell types. Metabolism carbohydrate, fatty acid *β* oxidation, glycolysis, OXPHOS, and TCA cycle exhibited a consistent trend between HDCs and DDCs, and all of them were up-regulated in the DDCs of DLH and CD-PC, while down-regulated in the DDCs of EnC, B cells, and Mac. Thereby, DLH and CD-PC but not EnC, B cells, and Mac may attribute to the role of substance transport within the kidney. These findings prove that the metabolism consisted with renal cell function and both of them showed a heterogeneity trend in DKD progress.

DEGs in the members of SLC superfamilies may explain the metabolic disorder in kidney (Appendix A). In the current study, we found that the expression levels of SLCs in mitochondria, including *Slc25a3*, *Slc25a4*, *Slc25a5* and *Slc25a30*, were significantly up-regulated in the DDCs of CD-PC (Appendix A) and might lead to an increase in ROS and enable the latter to act as a signaling molecule to activate tubular cell apoptosis in DKD [22]. The increase of ROS content and up-regulation of apoptosis could be found in both ROS staining (Figure 1C) and ssGSEA score (Figure 5E). In addition to mitochondrial SLCs, we found that professional carbohydrate and lipid SLCs were mostly up-regulated in the DDCs of DLH and CD-PC (Appendix A). The kidney has the highest resting metabolic rate in the body. The elevation of these SLCs is consistent with the observed elevation of ssGSEA score in the carbohydrate metabolism, gluconeogenesis, glycolysis, fatty acid oxidation, lipid metabolism, and fatty acid oxidation pathways in the DDCs of DLH and CD-PC (Figure 5E). A previous study has demonstrated that, even though the renal system intakes a large amount of glucose in DKD, the proportion utilized for aerobic oxidation is reduced instead. This shift towards the overactivated glycolysis breaks the metabolic balance, leading to oxidative stress and subsequent kidney damage [21]. Herein, we speculate that the oxidative stress damage of mitochondria may be caused by abnormal transport from the cytoplasm to mitochondria. 

The potential of SLCs as targets in treatment of DKD has been concerned by researchers. Invokana (inhibitor of Slc5a2) has been used for blood glucose management to alleviate kidney injury in type 2 diabetes patients. Slc5a2, encoding the SGLT2, is localized in the early proximal tubule and response for 90% glucose reuptake, while high glucose conditions increase SGLT2 levels and enhance glucose recovery in the proximal tubule of diabetic patients. By inhibiting the activity of SGLT2, blood glucose levels can be reduced, inflammation, fibrosis, and damage to the glomerular interstitium improved, and proteinuria reduced by 30% to 50% [23]. In the current study, we have observed that *Slc5a2* is upregulated in the DDCs of DLH compared to HDCs, while upregulation of *Slc2a4* in DDCs and CD-PC is also seen (Appendix A). *Slc2a4* is another SLC member that can transport glucose, and the knockout of this gene in podocytes can prevent glomerular hypertrophy, mesangial dilatation, albuminuria and other symptoms in mice induced by DKD, thus reducing the risk of DKD [24]. These members of the SLC superfamily could be potential treatment targets of DKD.

To reveal the characteristic genes associated with DDCs outside mitochondria, we have defined ESDEG with deleting MANGs. There are differences in RNA dynamics between DDCs and HDCs (Figure 5A–D). A massive single nuclear RNA-Seq (snRNA-Seq) study has documented that CD-PC, DLH, and EnCs exhibit the highest count of differentially expressed gene in DKD progression [25]. Furthermore, GO enrichment analysis of ESDEGs in these five cells revealed that they were belong to heterogeneous regulatory modes (Figure 6A). ESDEGs of DLH were enriched to wound healing, regulation of supramolecular fiber organization, cell-substrate adhesion, regulation of actin filament-based process etc., suggesting that it may have undergone cyto-dynamics changes. CD-PC have enriched abundant gene sets related to epithelial cell development or differentiation. B cells and Mac exhibited metabolism-related or immune-related pathways similar to the MHC pathways. Cell communication and TF prediction allow us to understand how these intricate pathways regulate complex biological network by controlling downstream TF and TG with a very small number of receptors (Figure 7). For instance, *Fzd7* codes a surface labeled molecule, and is expressed as a marker of progenitor cells or stem cells in mammalian renal mesenchymal [17]. As a receptor of ncWNT pathway, *Fzd7* was found to be significantly decreased in DDCs of CD-PC. It controlled the cell polarity of CD-PC through a total of 25 transcription factors such as EGR1, EP300, ESR1, ETV1, etc. (Figure 7B), resulting in the decreased developmental ability in the DDCs of CD-PC and showed a decreased proportion of G2M phase cells in the cell cycle. Consistent with this phenomenon, inhibition of Fzd7 tends to imply the decreased epithelialization and increased cell death and apoptosis [26]. *F11r* encodes junction adhesion molecular and plays a key role in the adhesion of endothelial and epithelial cells and is relevant to angiogenesis, immune response, and maintenance of vascular permeability. This gene ontology and TF regulatory network showed a similar trend in HDCs and the DDCs of CD-PC (Figure 7 and Figure 8). In the lesions of chronic diseases, irreversible cell proliferation stagnation, increased protein production, decreased anti-apoptotic ability and metabolic ability disorder often occur. For instance, impairment of cell cycle and proliferation can be found in renal tubular epithelial cells in CKD, HIV-associated nephropathy and polycystic nephropathy [27]. Evidence has demonstrated that hyperglycemia and lipotoxicity lead to tubular senescence in DKD in relation to cell death pathways such as apoptosis, autophagy, necrosis and ferroptosis [28]. In the current study, data have showed that in the DDCs of CD-PC, there was a state of Epi development and cell cycle accompanied by the decreased expression of *Fzd*, which not only corresponded to the impaired normal function of CD-PC, but also duplicated the impaired metal ion transport ability of CD-PC in the process of DKD [10]. What excites us is the proportion shift of DDCs and HDCs in CD-PC can not only be testified between Bulk RNA-Seq deconvolution of DM and DKD (Figure 8G), but also be reversed by treating with ARB and HKC. However, since SGLT2i acts on S1 PTCs, we did not find that it has a reverse effect on the composition of DDCs and HDCs in CD-PC (Figure 8D).

The metabolic and cell cycle changes described above may further cause fibrosis and inflammation of renal tubules. The previous study has demonstrated that M1-type Mac play a leading role in DKD, and T cells may have an extensive synergistic effect with Mac in DKD inflammation [10]. In the current study, we have alternately provided evidence suggesting that B cells, except T cells, may also be driven by mitochondria and participate in the process of DKD with *Cd74* in the APP pathway (Figure 7A,B) and its interaction with other renal cells through MHC-II, TGF-*β*, in the SELL pathway (Table 2 and Appendix A). *Cd74* encodes a class of MHC-Ⅱ transmembrane glycoprotein molecules and plays a certain role in the regulation of macrophage migration and T and B cell development. *Cd74* receptor activity was first discovered in B cells, in which, this gene regulates cell cycle and immune activation of B cells [29]. As an additional survival receptor for B cells, the activation of Cd74 can induce B cells to enter the S-phase to improve their anti-apoptotic ability and thus enhance their survival ability, which leads to the formation of B cell libraries and the activation of immune response [19]. Therefore, the activation of Cd74 is associated with plentiful inflammatory diseases, such as fibrosis, T1D and systemic lupus erythematosus [30]. In the kidney, the absence of this gene is usually a significant obstacle to glomerular injury, nephritis and renal tubule fibrosis [30]. Intriguingly, the up-regulation of B cell DDCs caused by DKD can be reversed by SLGT2i, ARB and HKC.

There are limitations in the current study. The cells in glomerulus, including podocytes and mesangial cell cannot be captured by scRNA-Seq, but these cells are important in renal injury and cause of proteinuria in DKD. This is due to a combination of the peculiarity of 10X Genomics and the small number of glomerular cells in total renal cell number. Another disadvantage of scRNA-seq is that the spatial coordinate information is lost during the process of tissue dissociation into single-cells suspension. Further investigation of the transcriptome map in DKD with spatial information has been taken into our consideration.

In conclusion, the current study has provided experimental evidence revealing the heterogeneity of mitochondrial disorders in distinct cell types, particularly in the CD-PC and B cells during the process of DKD and administration of ARB, SGLT2i and HKC. The information is useful for the discovery of new biomarkers in DKD and the proposal of diagnosis and treatment schemes for the disease.

## 4. Methods and Materials

### 4.1. Animal Management

BKS.Cg-Dock7m ^+/+^ Lepr^db/J^ (db/db) mice (DKD group) and heterozygous db/m mice (Ctrl group) aged 10 weeks were acquired from Huachang Xinnuo Medical Technology Co., LTD (Taizhou, China). All experiments with mice were carried out in accordance with the guidelines of, and were approved by, the Institutional Animal Care and Use Committee at China Pharmaceutical University (CPU). All male mice were housed in cages with a barrier environment (22–25 °C; 40–50% humidity; and 12-h light/dark cycle) and received regular chow and water at liberty. At 16 weeks, UACR and BG were measured as we described before [10]. Based on blood glucose and UACR results, mice in the DM (BG > 16.7 mmol/L) and DKD (UACR > 30 ng/μg) groups were identified and four left kidneys of each group were used for bulk RNA-Seq. At 20 weeks, the mice were sacrificed by cervical dislocation method and the peripheral blood was removed by cardiac perfusion with 1x phosphate buffered solution and the kidneys of both sides were collected. The left kidney was placed in a frozen tube and stored in liquid nitrogen, and the right kidney was uniformly cut lengthwise and placed in 4% paraformaldehyde solution and 4% glutaraldehyde solution, respectively.

### 4.2. Reactive Oxygen Species, Hematoxylin-Eosin, and Periodic Acid-Schiff Stain

After the kidney was removed from the cryostorage tube, the frozen section was placed in the Cryotome E freezing microtome (Thermo, Shanghai China). The fluorescence staining and sealing were then performed in accordance with the specification of the D7008 ROS kit (Servicebio, Wuhan, China). Images were collected by Fluorescent Microscopy (NIKON ECLIPSE C1, NIKON). DAPI glows blue at a UV excitation wavelength of 330–380 nm and emission wavelength of 420 nm; FITC glows green at an excitation wavelength of 465–495 nm and emission wavelength of 515–555 nm; CY3 glows red at an excitation wavelength 510–560 nm and emission wavelength of 590 nm. In the Hematoxylin-Eosin (HE)-staining process, half of the right kidneys were embedded in paraffin wax, and then dehydration, staining, and elution were performed in accordance with the specification of the HE-staining kit (Hunan Aifang Biotechnology CO., Ltd., Changsha, China). Finally, the slices were sealed with neutral gum and the images were collected by AE41 optical microscope (Motic, Shenyang, China). In the PAS-staining procedure, the section and dewatering process after paraffin embedding was consistent with the HE procedure. The actual use of the above PAS process was included in the G1008 PAS dye kit (Hunan Aifang Biotechnology CO., Ltd., Changsha, China).

### 4.3. Transmission Electron Microscopy

The upper part of the kidney was taken from the glutaraldehyde and rinsed with 0.1 M phosphoric acid buffer 3 times for 15 min each time. The sample was infiltrated with 1% osmic acid solution configured with 0.1 M phosphoric acid buffer to avoid light for 7 h, and then the phosphoric acid rinsing operation was repeated. Alcohol gradient dehydration for 1 h each: 30%, 50%, 70%, 80%, 95%, 100%, 100%. Different proportions of the mixture of anhydrous ethanol and acetone were used for dehydration: 3:1, 0.5 h; 1:1, 0.5 h; 1:3, 1.5 h; 0:1,1 h. The samples were treated at 37 °C for 12 h with 812 embedding agent and then treated at 60 °C for 48 h to complete polymerization. The ultrathin sectioning mechanism was used to make sections ranging from 60 to 80 nm. After the dyeing step, the images were collected: 2% uranium acetate saturated alcohol solution away from light, 8 min; 70% alcohol rinse, 3 times; ultra-pure water rinse, 3 times; 2.6% lead citrate solution away from carbon dioxide, 8 min; ultra-pure water rinse, 3 times; the filter paper absorbs excess water and dries at room temperature for 12 h.

### 4.4. ScRNA-Seq Data Preprocessing, Dimension Reduction, Clustering, Cell Annotation and Quality Control

Renal scRNA-seq data of 4 Ctrl mice were adopted from the National Center for Biotechnology Information (NCBI) GEO database (GSE107585) and scRNA-Seq data of DKD kidneys were previously registered in Sequence Read Archive (PRJNA749372). The steps of quantitative processing, quality control, filtering, clustering, and annotation after obtaining FASTQ files of scRNA-Seq were described in our previous publications [10]. Seurat container with cell annotation information was generated, and a total of 58,259 individual cells were obtained. After the cell annotation was completed, the DoubletFinder (Version 2.0.3) package was used to classify the doublets and singlets by iterating the optimal parameter of proportion of artificial nearest neighbors (pANN) based on principal component (PC) domain size as 1 to 30. We used the find.pK() function of DoubleFinder package to determine the optimal pANN value of proximal tubular cells (PTC) and rest cell types as 0.1 and 0.18 respectively, and relied on the pANN value to identify double or multiple droplets (doublets) and single droplets (singlets) under Poisson distribution. 

As described in the methods above, the uniform manifold approximation and projection (UMAP) of single-cell droplet identification of PTC and other cell types are shown in Appendix A, respectively. The proportion of doublets and singlets within all PTCs and the remaining cell types are visualized in Appendix A and the count data are shown in Appendix A. We noticed that PTCs contained abundant doublets (21.7%, 9804/45,205). Among the PTC types, S2 (2.86%, 696/24,347) had the least proportion of doublets and S3 (50.8%, 6287/12,385) had the greatest proportion of doublets (Appendix A). The Double-finder was less sensitive when used for unsupervised cell clusters with similar expression patterns [31]. We thus realized that the false positive heterotypic doublets in S1 and S3 might be incorrectly identified. Among the DRSC, we found that Mac (62.7%, 260/415) had the highest percentage of doublets, while the ALH (0.0341%, 1/2932) had the lowest proportion of doublets (Appendix A). Considering these factors, we refused to discard doublets misidentified in PTCs but did not reserve doublets in other cells for downstream advanced analysis. Consequently, we used the Subset function in the Seurat (Version 4.0.3) package to take out all cells in PTC and singlets of other cells for subsequent analysis. The RunUMAP function in Seurat was re-executed and UMAP score were visualized by adding confidence intervals via ggplot2 (Version 3.3.5).

### 4.5. Gene Set Scoring and Enrichment Analysis

The calculation of mitochondrial coding gene set content was performed by the PercentageFeatureSet() function in Seurat. After the percentage was obtained, the wilcox.test() in R platform was used to calculate the difference. The normalized expression quantity of each cell was used as GSVA (Version 1.44.4) input matrix and Kernel estimation of the cumulative density function (kcdf) was calculated according to Gaussian distribution to obtain GSVA and single sample ssGSEA scores. The inter-group difference analysis of GSVA and ssGSEA scores was implemented by the limma (Version 3.48.3) package in R based on empirical linear models of Bayesian methods. The *p* value of both differential analysis is corrected by false discovery rate (FDR) to get the adjusted *p* value. Then, GSVA significant pathways (*p* adjustment < 0.05 and |Log2FC| > 0.5) were selected. Linear regressions between DEGs were calculated by lm() function based on the qr method and visualized by ggplot2. 

In the current study, DCP, GO and KEGG enrichment analyses were implemented in ClusterProfiler (Version 4.4.4) [32] through the principle of hypergeometric distribution. In DCP enrichment analysis, among 518 intersection genes belonging to DLH, 297 were enriched in 33 DCP (9 pathways met the significant enrichment condition of *p* < 0.05), such as the Glutathione metabolism, Taurine and hypotaurine metabolism, ROS elimination, Biosynthesis of unsaturated fatty acids, Alanine aspartate and glutamate metabolism and so on. While among 324 exclusive genes of DA2 in DLH, 140 of them were enriched in 27 DCP and only one pathway, named Collagen formation, was significantly enriched. Among 664 intersection genes belonging to CD-PC, 112 were enriched in 28 DCP (2 of them met the significant condition), such as Glutathione metabolism, ROS elimination, EMT, Glycolysis Gluconeogenesis and Apoptosis. While among 218 exclusive genes of DA2 in CD-PC, 80 of them were enriched in 16 DCP and only TGF-*β*1 pathway was significantly enriched. Among 115 intersection genes belonging to EnCs, 26 were enriched in 13 DCP (ROS-GSEA and Glutathione metabolism met the significant condition). Among 510 exclusive genes of DA2 in EnCs, 145 of them were enriched in 29 DCP; however, only the Glycerolipid metabolism and Glutathione metabolism pathways were significantly enriched. Considering B cells, 76 of 297 intersection genes were enriched in 19 DCP (ROS GSEA and TGF-β pathways were significantly enriched), while 245 of 563 exclusive genes of DA2 were enriched in 25 DCP (Glutathione metabolism, Alanine, aspartate and glutamate metabolism, ROS elimination, Nitrogen metabolism and Arginine and proline metabolism pathways were enriched). Regarding Mac, 80 of 146 intersection genes were enriched in 20 DCP (Apoptosis and ROS GSEA pathways were significant). While 146 of 835 exclusive genes of DA2 were enriched in 39 DCP, however, only Glycine, serine and threonine metabolism pathway were significantly enriched (Appendix A).

In the pseudotime analysis, we first took out the UMI count in the Seurat object that conforms to the negative binomial distribution to create the CellDataSet object of monocle (Version 2.18.0). DDRTree dimensionality reduction determined the topological structure of data, and we then used exclusive significant difference genes (ESDEG) of corresponding cells to guide the semi-supervised trajectory analysis while the orderCells() function was used to determine the coordinates of the cells in pseudotime [33]. Trajectory diagram and its color rendering was performed by the plot_cell_trajectory() function in monocle, while at the same time, the cumulative density distribution curve generated by ggridges (0.5.3) and *t* test was executed by ggsignif (0.6.3).

### 4.6. Cell Communication Prediction

After the normalized matrix of the DKD group in the Seurat object container is obtained, the input RNA matrix was converted into a reliable protein network with STRINGdb support using a random propagation technique through projectData() function. Then, by referring to the ligand–receptor interactions database containing 229 signaling pathways including secreted signaling, cell–cell contact, ECM–Receptor three types in CellChat (Version 1.5.0), the obtained protein matrix was calculated by computeCommunProbPathway() function to obtain the intercellular interaction intensity to represent the possibility of manifold-leaning inferred intercellular communication network. In order to focus on the heterogeneity of cell subsets within DKD, five cell types including DLH, CD-PC, EnCs, B cells and Mac, were divided according to HDC and DDC, and the rest of the cells were not. Considering that some subsets contained only a tiny number of cells, we did not set the minimum number of cells involved in the pathway for filtration. Consequently, 46 pathways including 90 ligand receptor pairs verified by permutation test (*p*-value < 0.05) were preserved. The downstream visualization of cell communication was performed through netVisual_chord_cell (), netAnalysis_signalingRole_scatter (), netAnalysis_signalingRole_heatmap (), netVisual _bubble () and so on provided by CellChat. 

### 4.7. Transcription Factor Prediction

The single-cell transcription factor regulatory network is mainly completed by SCENIC [16] (Version 1.3.1) and its dependent GENIE3 [34] (Version 1.18.0), RcisTarget (Version 1.3.1) and AUCell [16] (Version 1.1.6) packages in R. To avoid artificial co-variation, summarized counts (UMI count) were used as input data of GENIE3 (R version) to calculate the co-expression between TF and potential target gene through random forest models. By using motif framework of iRegulon information recorded by i-cisTarget (mm9-500bp-upstream-7species.mc9nr.feather and mm9-tss-centered-10kb-7species.mc9nr.feather), TF-motif enrichment analysis was carried out on sequences of 10 kb around and 500 bp upstream from the transcriptional start site (TSS). In the results, a normalized enrichment score (NES) greater than 3.0 was regarded as a qualified regulon and preserved. The genetic matrix of regulon was scored by AUCell packages, a method for area under the curve (AUC) calculation. According to the correspondence between TF and TG in RcisTarget and hypotaxis of receptors and TF in scMLnet [35], we constructed the regulatory network of receptor, TF and TG. The continuous AUCell score of regulon was analyzed for difference through the limma package, Then circlize [36] (Version 0.4.15) and plotly (Version 4.10.0) were performed, respectively, to display top50 (if sufficient) significantly different regulon (*p* value < 0.01) in annular heat map and sankey diagram.

### 4.8. Bulk RNA-Seq Was Deconvolved by Cibersort Using the scRNA-Seq Data

Fastq files of bulk RNA-Seq were trimmed and aligned to the reference index (mm10 Version). Then gene counts were obtained by featureCount (Version 2.0.1) to calculate the transcripts per million (TPM). After figuring out the top 10 marker genes of 10 sub cell cluster, deconvolution was implemented by Cibersort.

### 4.9. scRNA-Seq Labels Transformation

In addition to the Ctrl and DKD groups, scRNA data from the HKC treatment group came from SRA (PRJNA991651), data of ARB and Sodium-glucose cotransporter-2 inhibitors (SGLT2i) treatment groups came from GEO (GSE181382). The data of all the groups underwent the same preprocessing process and were annotated with the same cell type as the DKD group. Finally, CD-PC and B cells of the Ctrl, HKC, ARB and SGLT2i groups were severally transformed with labels of HDC and DDC through singleR (Version 1.10.0).

## Figures and Tables

**Figure 1 ijms-24-13502-f001:**
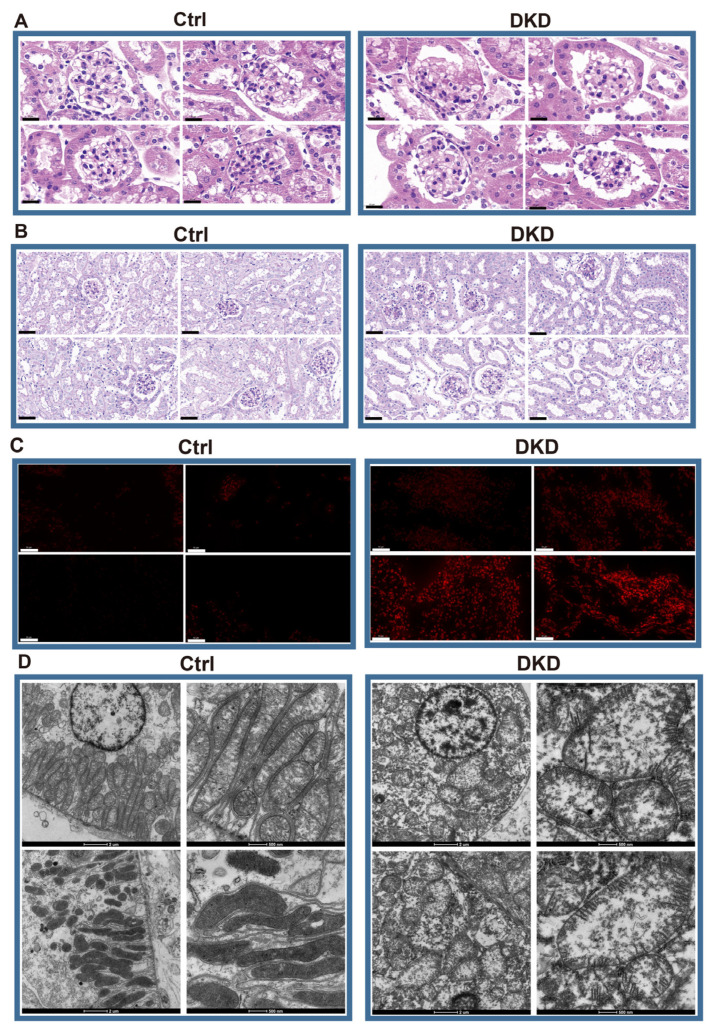
Histopathological examination showed that there were significant differences between DKD group and Ctrl group in glomerular injury, renal tubule lesions, ROS overproduction and mitochondrial morphology. (**A**) HE optical photograph of DKD group kidneys showed thickening and diffuse hyperplasia of glomerular basement membrane, early Kimmelstiel–Wilson (KW) nodule formation, glomerular atrophy, transparent capillary lesions. Scale bar, 20 μm. (**B**) PAS results showed that DKD group developed obvious lesions such as necrosis and exfoliation of renal tubular epithelial cells, vacuolar lesions and dilatation of renal tubular lumen. Scale bar, 80 μm. (**C**) By 2,7-Dichlorodihydrofluorescein diacetate dyeing (red), it can be obviously observed that ROS levels in the frozen kidney section of the DKD group began to rise. Scale bar, 50 μm. (**D**) The morphological deterioration of DKD mitochondria was observed under electron microscope. Scale bar, 2 μm and 500 nm.

**Figure 2 ijms-24-13502-f002:**
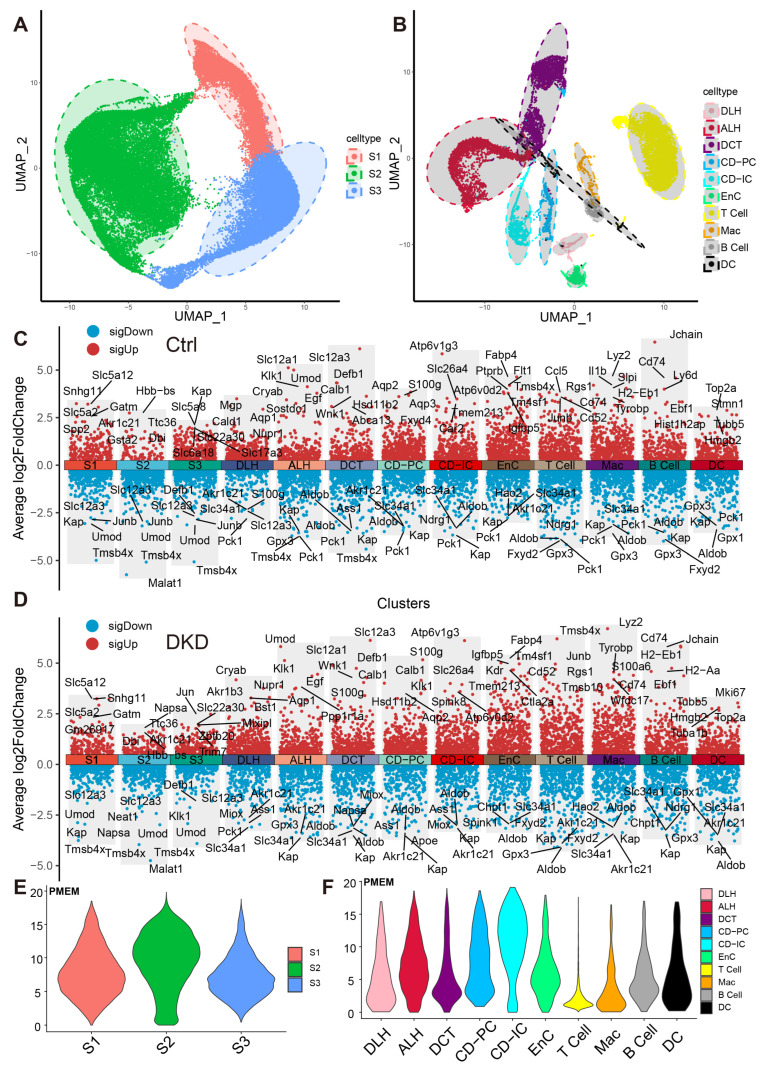
Dimension reduction and mitochondrial content calculation of scRNA-Seq. (**A**,**B**) PTCs in S1, S2 and S3 segments of PTCs (**A**) and DRSC (**B**), including ALH, CD-IC, CD-PC, DCT, DLH, EnC, T cell, B cell, Mac and DC, are respectively shown in the UMAP map. The accompanying shadows indicate confidence intervals of UMAP score. (**C**,**D**) The combined scatter plot shows the respective markers of the 13 renal cell types in the Ctrl (**C**) and DKD (**D**) groups. The fold change was obtained by comparing the corresponding cells with the remaining cells except the cells themselves and the five marker genes with the lowest *p*-value (Wilcox test) in each cell of both up-regulation and down-regulation were labeled. E–F: The PMEM of PTCs (**E**), DRSC and IC (**F**) were demonstrated in violin plots with consistent y-coordinates. ALH—Ascending loop of Henle; CD-IC—Collecting duct intercalated cell; CD-PC—Collecting duct principal cell; DCT—Distal convoluted tubule; DLH—Descending loop of Henle; EnC—Endothelial cell; Mac—Macrophage; DC—Dendric cell; PTCs—Proximal tubule cells; S1, S2 and S3 are segments of proximal tubule; PMEM—percentage of mitochondrial encoding mRNA; DRSC—distal renal stromal cells.

**Figure 3 ijms-24-13502-f003:**
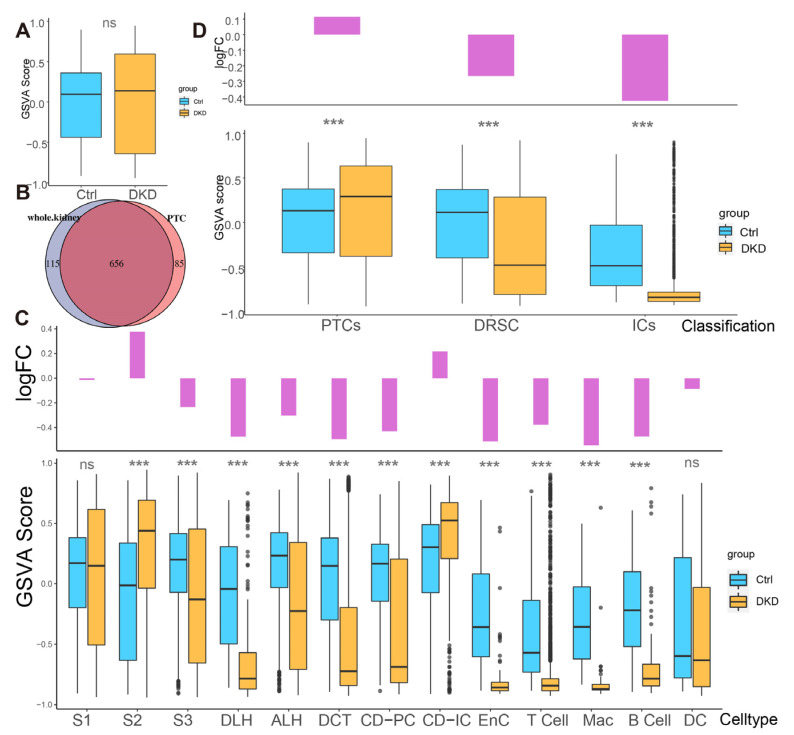
Different analysis of mitochondrial mRNA content between the Ctrl and DKD group at different classification level s. (**A**) The boxplot shows the mitochondrial encoding gene GSVA score of all individual renal cells in the DKD (yellow) and Ctrl (blue) groups. (**B**) There was a large degree of overlap between the comparison of DKD and Ctrl with the whole kidney as the unit and the comparison between groups of separate PTCs. (**C**) Bar-boxplot composite diagram depicting the logarithmic fold change of GSVA score (bar plot) of the DKD group compared to the Ctrl group and distribution of GSVA score in both groups (boxplot). (**D**) Bar-boxplot composite diagram depicting the logarithmic fold change of GSVA score (bar plot) of PTCs, DRSC, ICs in the DKD group compared to that in the Ctrl group and distribution of GSVA score in both groups (boxplot). *** *p* < 0.001; ns: No significance.

**Figure 4 ijms-24-13502-f004:**
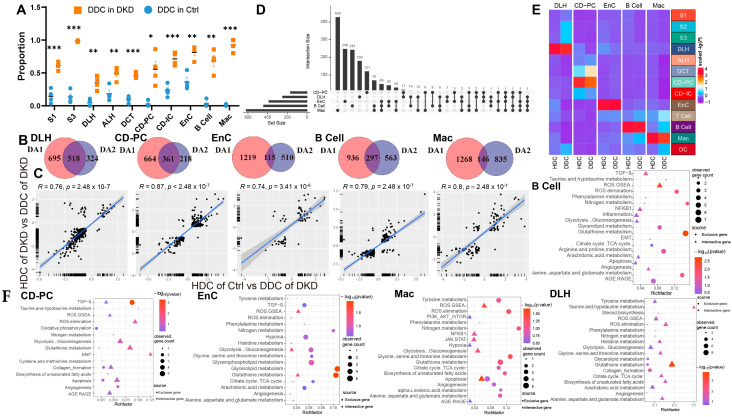
Cells of the DKD group can be divided into two distinct clusters by the clustering of mitochondria-related gene sets. (**A**) Boxplot showing the distribution in DKD-dominance clusters among cell types of the Ctrl group (skyblue) and DKD group (orange). P, unpaired *t* test. * *p* < 0.05, ** *p* < 0.01, *** *p* < 0.001. (**B**) Venn diagram showing the difference (*p* < 0.05 and |log_2_(fold change)| > 0.5) between HDC in the Ctrl (DA1) and DKD (DA2) groups and DDC. (**C**) The linear relationship of the log_2_ (fold change) of intersected genes in (**C**) was fitted by Pearson correlation coefficient. (**D**) High latitude Venn diagrams of DA2 unique genes among DLH, CD-PC, EnC, B cells, Mac. (**E**) The correlation between sub-cell types and their original cell types is displayed by complex heatmap. (**F**) Bubble diagram of conventional DKD-related pathway enrichment analysis of diff2-exclutive genes (Blue area), intersective genes (Purple area) of DA1 and DA2 in Figure 3C. The top 20 pathways (if sufficient) were selected by *p*-value (<0.05).

**Figure 5 ijms-24-13502-f005:**
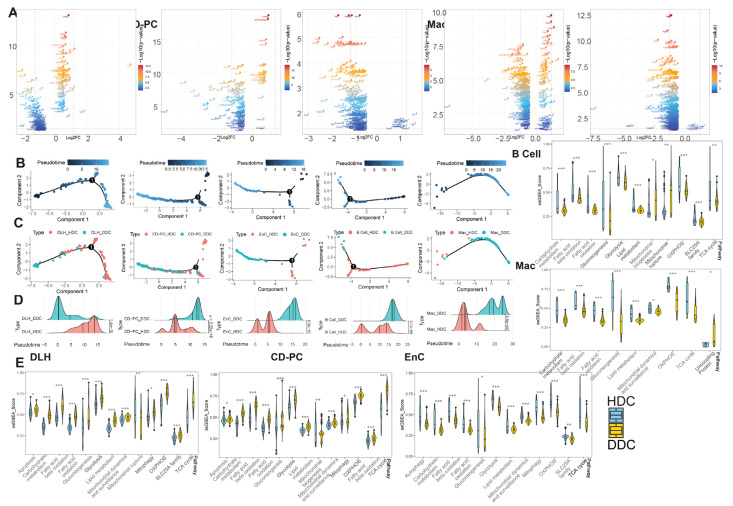
The pseudotime analysis of ESDEG and mitochondria-related DKD pathways demonstrated that HDCs and DDCs were significantly different in cell state. (**A**) The volcano diagram shows the differences of exclusive genes (The blue area in Figure 3C) between DDCs and HDCs in DLH, CD-PC, EnC, B cells, Mac, where the abscess represents the difference fold and the ordinate represents the significance (Q value < 0.05, |log_2_(Fold Change)| > 0.5). (**B**,**C**) The scatter plot shows the pseudotime distribution trajectory (**B**) obtained by semi-supervised learning in monocle2 and the coordinates (**C**) of HDCs and DDCs in each cell in the pseudotime. (**D**) The cumulative density distribution curves show that HDCs and DDCs have significant differences in the pseudotime distribution of their respective cells (*t* test). (**E**) The gene set enrichment analysis (ssGSEA) score of DKD-related mitochondrial pathways related to mitochondrial damage in DKD is demonstrated in HDCs and DDCs, and the difference analysis was performed by generalized linear models in the limma package. (*, *p*-value < 0.05; **, *p*-value < 0.01; ***, *p*-value < 0.001).

**Figure 6 ijms-24-13502-f006:**
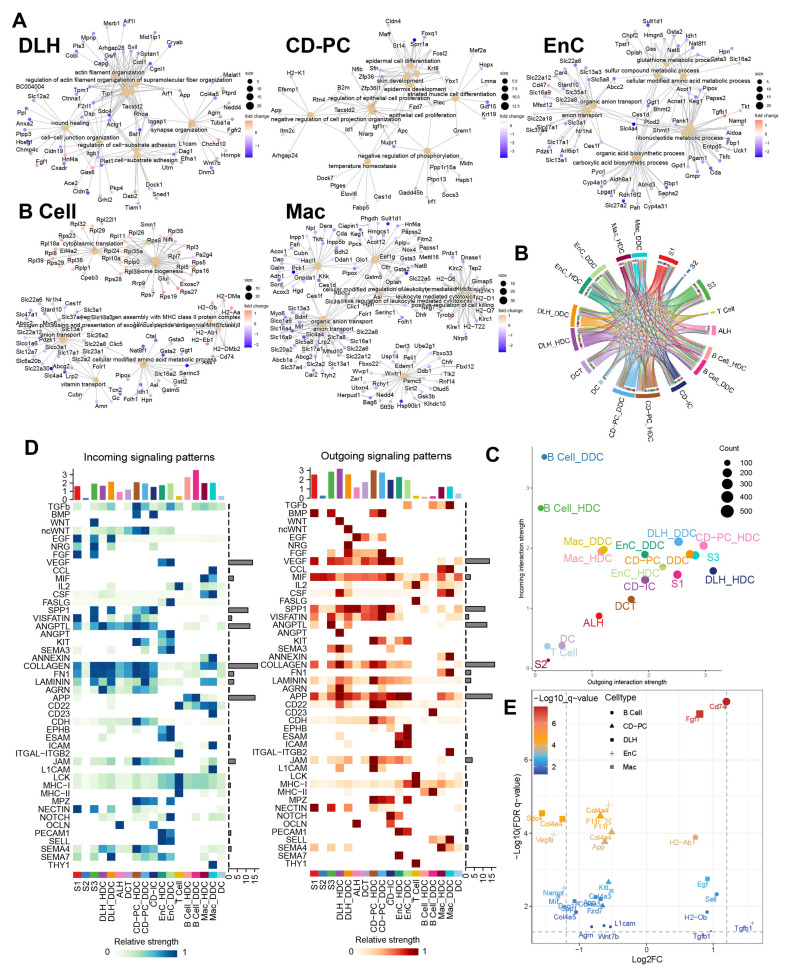
Gene ontology enrichment analysis of ESDEGs and prediction of the cell communication pathway. (**A**) Gene ontology enrichment analysis (based on the principle of hypergeometric distribution) network diagram of ESDEGs produced by clusterProfiler and CNBplot. The color of the outer dot represents the foldchange of ESDEGs in DDCs compared to that in HDCs, while the size of the inner dot represents the number of genes enriched into this pathway. (**B**) Comprehensive cell communication of each cell type in the DKD group was obtained by Cellchat package calculation and its built-in ligand-receptor information. The color of the string represents the type of cell from which the communication pathway originated, and the length of the arc represents the number of cells involved in the corresponding pathway. (**C**) Each cell type in the DKD group was involved in the incoming and outgoing cell communication to varying degrees. The size of the bubble represents the actual number of cells involved in communication. (**D**) The two heat maps respectively showed the relative strength of 46 incoming (left) and outgoing (right) cell interactions of each cell type in the DKD group. The bar chart at the top showed the total relative strength of the corresponding cell type in each of the 46 signaling pathways. (**E**) The volcano diagram showed the q value and fold change of ESDEGs as ligand or receptor for DDCs of B cells, CD-PC, DLH, EnCs and Mac compared to HDCs included in cell communication pathways.

**Figure 7 ijms-24-13502-f007:**
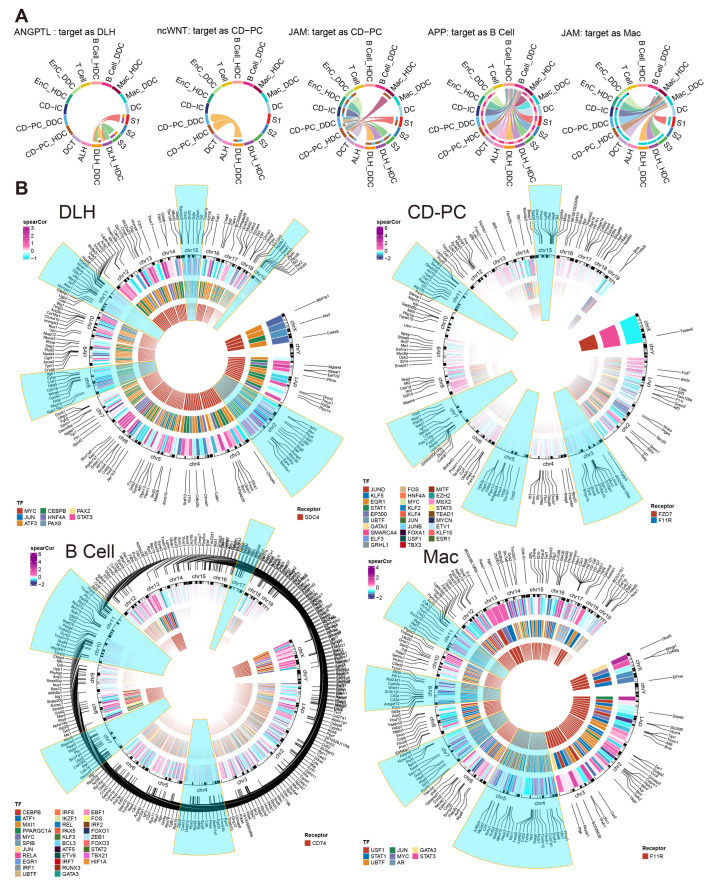
Incoming cell communication pathways, including ESDEGs between DDCs and HDCs of DLH, CD-PC, B cells and Mac, which regulate downstream TF and TG. (**A**) Chords of five incoming cell communication pathways containing corresponding cell receptors as ESDEGs. The color of the string represents the type of cell from which the communication pathway originated, and the length of the arc represents the number of cells involved in the corresponding pathway. (**B**) The three-layer ring heat map shows the co-expression coefficient (Spearman) between target gene and transcription factor, transcription factor and receptor from the outside to the inside. The outermost content marks the location of ESDEGs in the mouse genome, and the five chromosomes that contain the most ESDEGs are highlighted in blue (if sufficient).

**Figure 8 ijms-24-13502-f008:**
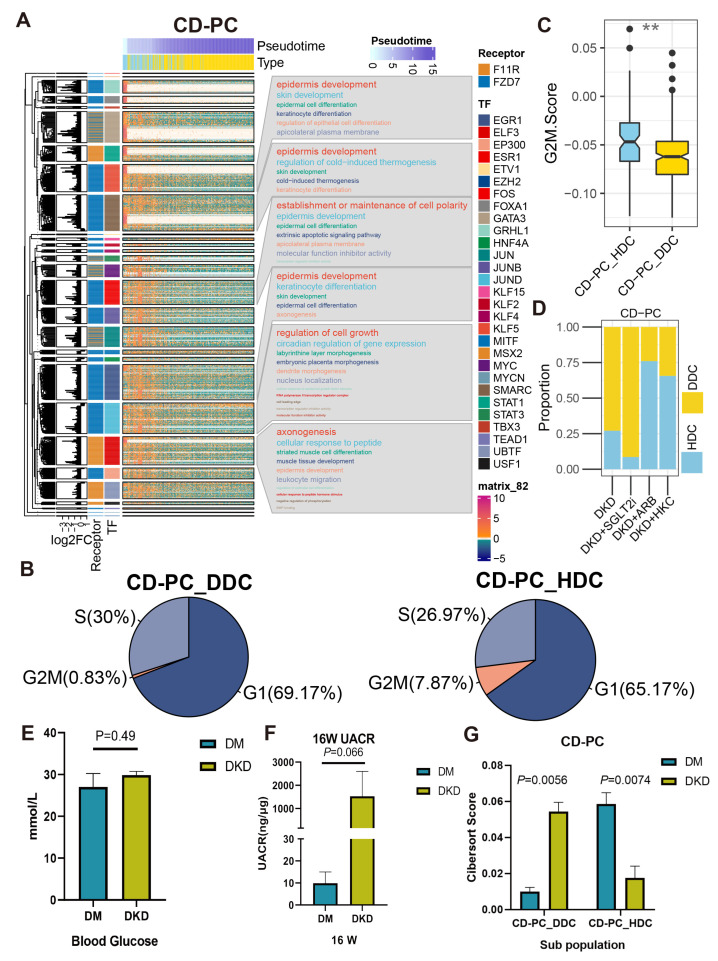
The cascaded regulatory network of receptors, TF and TG resulted in the regulation of specific KEGG pathways and cell state of HDCs and DDCs. (**A**) The complex heat map showed the distribution of HDCs and DDCs of CD-PC as well as the expression of ESDEG with the progression of pseudotime. The four columns of left annotation from left to right are kmean clustering tree, the fold change of ESDEG to HDC in DDC, receptor and TF, respectively. More than 50 genes belonging to the same transcription factor will be enabled for KEGG enrichment analysis and shown in the word cloud annotation on the right. (**B**) Pie charts of cell cycle distribution of two subpopulations of CD-PC in the DKD group. The G2M phase cells of CD-PC_HDC were significantly higher than those of CD-PC_DDC. (**C**) The G2M score calculated by cell cycle signatures was significantly higher in CD-PC_HDC than in CD-PC_DDC. The S-phase score calculated by cell cycle signatures was significantly higher in B cell_DDC than in B cell_HDC. (**, *p* < 0.01, *t* test). (**D**) The distribution of HDCs and DDCs in B cells in the ARB, SGLT2i and HKC groups was predicted by singleR. (**E**,**F**) At the 16th week, db/db mice were used to measure BG (**E**) and UACR (**F**). Mice with high BG (>16.7 mmol/L) but no proteinuria (UACR < 30 ng/μg) were divided into the DM group, and mice with high blood glucose (BG) and proteinuria (UACR > 30 ng/μg) were divided into the DKD group. (**G**) Cibersort score of HDCs and DDCs of CD-PC inferred by TPM value of bulk RNA-Seq.

**Table 1 ijms-24-13502-t001:** Parameters in dimension reduction cluster based on GSVA score.

Cell Type	DEP	Dims of PCA	Resolution
S1	138	20	0.1
S2	127	19	0.08
S3	142	20	0.1
DLH	67	3	0.15
ALH	119	16	0.1
DCT	129	15	0.1
CD-PC	125	8	0.1
CD-IC	104	9	0.2
EnC	44	3	0.15
T Cell	86	10	0.1
B cell	32	2	0.1
Mac	76	3	0.1
DC	10	NA	NA

**Table 2 ijms-24-13502-t002:** Detailed list of SLC genes as cell markers.

Gene	As Marker in	Substrate
*Slc12a1*	ALH	sodium-potassium-chloride
*Slc16a7*	DCT	lactate, pyruvate, ketone bodies
*Slc26a4*	CD-IC	sulfate
*Slc43a2*	CD-IC	L-isomers of neutral amino acids, including leucine, phenylalanine, valine and methionine
*Slc4a9*	CD-IC	anion
*Slc8a1*	CD-PC	Ca^2+^
*Slc12a3*	DCT	Sodium, chloride
*Slc27a2*	PTC	free long-chain fatty acids
*Slc34a1*	PTC	sodium-phosphate
*Slc13a3*	S3	succinate and other Krebs cycle intermediates
*Slc17a1*	S1, S3	-
*Slc17a3*	S3	intracellular urate and organic anions
*Slc22a30*	S3	-
*Slc22a1*	S1, S3	organic cation
*Slc22a12*	S3	urate
*Slc22a8*	S1, S3	organic anions
*Slc23a1*	S1, S3	vitamin C
*Slc2a2*	S1, S3	glucose
*Slc6a20b*	S1, S3	-
*Slc37a4*	S1, S3	glucose-6-phosphate
*Slc3a1*	S1, S3	neutral and basic amino acids
*Slc47a1*	S1, S3	-
*Slc4a4*	S1, S3	bicarbonate
*Slc6a19*	S1	neutral amino acids
*Slc5a12*	S1	lactate
*Slc5a2*	S1	sodium, glucose
*Slc7a7*	S1	cationic amino acid, neutral amino acids
*Slc7a8*	S1	-
*Slc22a6*	S3	sodium, organic anions
*Slc5a10*	S3	sodium, glucose, ascorbate, choline, iodide, lipoate, monocaroboxylates, pantothenate
*Slc6a18*	S3	sodium, neurotransmitters, amino acids, and osmolytes (eg., betaine, taurine, and creatine).
*Slc5a8*	S3	lactate, monocarboxylates, short-chain fatty acids, sodium
*Slc43a3*	EnC	-
*Slc9a3r2*	EnC	sodium, hydrogen

**Table 3 ijms-24-13502-t003:** SLC25 Family list.

Gene	As DEG in	Substrate
*Slc25a25*	DLH, Mac	ATP, Ca^2+^
*Slc25a5*	DLH, CD-PC, EnC, B cell	adenine nucleotide
* Slc25a51 *	DLH	NAD
*Slc25a16*	DLH, B cell, EnC	nucleotide
* Slc25a3 *	DLH, CD-PC, EnC	phosphate
* Slc25a15 *	B cell, EnC	L-arginine, L-lysine, L-ornithine
* Slc25a4 *	B cell, CD-PC	ATP/ADP antiporter
* Slc25a42 *	DLH	adenylic acid, coenzyme A
* Slc25a30 *	CD-PC	C4-dicarboxylate, sulfur compound
* Slc25a39 *	DLH, B cell, CD-PC, Mac	glutathione
* Slc25a10 *	DLH, B cell, Mac	dicarboxylic acid, malate, oxaloacetate, phosphate ion, succinate, sulfate

**Table 4 ijms-24-13502-t004:** DEG of SLC superfamilies (except SLC25) in HDCs and DDCs of DLH, CD-PC, EnC, B cells, and Mac.

Gene	As DEG in	Substrate
* Slc27a2 *	B cell, EnC, Mac	long-chain fatty acids
* Slc6a18 *	DLH	Sodium, neurotransmitters, amino acids, osmolytes (e.g., betaine, taurine, and creatine).
* Slc34a1 *	DLH, CD-PC, EnC, B cell, Mac	sodium-phosphate
* Slc4a4 *	DLH, B cell. EnC, Mac	bicarbonate
* Slc7a13 *	DLH	L-cystine, L-glutamate, aspartate
* Slc23a1 *	DLH, B cell	vitamin C
* Slc3a1 *	DLH, EnC, B cell	neutral and basic amino acids
* Slc7a12 *	DLH	amino acid
* Slc47a1 *	DLH, EnC, B cell	L-arginine
* Slc17a1 *	DLH, EnC, B cell, Mac	sialic acid
* Slc5a8 *	DLH, EnC, Mac	lactate, monocarboxylates, short-chain fatty acids, sodium
* Slc13a1 *	DLH, EnC, B cell. Mac	sodium-sulfate symporter
* Slc9a3r1 *	DLH	dopamine receptor, phosphatase
* Slc16a9 *	DLH, EnC, Mac	carnitine, monocarboxylic acid, creatine
* Slc12a3 *	DLH, CD-PC,	sodium, chloride
* Slc22a12 *	DLH, EnC, B cell, Mac	urate
* Slc6a13 *	DLH	taurine, amino acid, creatine, gamma-aminobutyric acid, monocarboxylic acid, neurotransmitter
* Slc5a10 *	DLH, EnC, Mac	sodium, glucose, ascorbate, choline, iodide, lipoate, pantothenate
* Slc22a1 *	DLH, EnC, B cell	organic cation
* Slc22a18 *	DLH, EnC	organic anion, ubiquitin protein ligase
* Slc10a2 *	DLH	bile acid-sodium
* Slc5a2 *	DLH, B cell, Mac	Sodium, glucose
* Slco3a1 *	DLH, B cell	Oligopeptide, sodium-independent organic anion, prostaglandin
* Slc17a3 *	DLH	intracellular urate and organic anions
* Slc5a12 *	DLH, B cell	lactate
* Slc2a5 *	DLH, Mac	fructose, glucose
* Slc22a13 *	DLH	nicotinate, urate
* Slco4c1 *	DLH	sodium-independent organic anion
* Slc2a2 *	DLH, B cell, Mac	glucose
* Slc6a20b *	DLH, B cell	L-proline
* Slc15a2 *	DLH	dipeptide, oligopeptide
* Slc29a3 *	DLH, Mac	nucleoside
* Slc22a6 *	DLH, EnC, B cell, Mac	sodium, organic anions
* Slc22a30 *	DLH, B cell	short-chain fatty acid
* Slc38a3 *	DLH	L-glutamine, L-histidine
* Slc16a2 *	DLH, EnC B cell	monocarboxylic acid, thyroid hormone
* Slc16a4 *	DLH, Mac	monocarboxylic acid
* Slc51b *	DLH	bile acid
* Slc23a3 *	DLH	vitamin C, sodium
* Slco1a6 *	DLH, B cell, Mac	bile acid, sodium-independent organic anion
* Slc38a2 *	DLH	L-glutamine, L-serine
* Slco4a1 *	DLH	sodium-independent organic anion, thyroid hormone
* Slc22a23 *	DLH	NA
* Slc12a2 *	DLH	K^+^, Hsp90, ammonium, sodium-potassium-chloride
* Slc4a7 *	B cell	sodium-bicarbonate symporter
* Slc30a9 *	B cell, Mac	monatomic cation, nuclear receptor
* Slc6a19 *	B cell, Mac	neutral amino acids
* Slc37a4 *	EnC, B cell, Mac	glucose-6-phosphate
* Slc22a8 *	B cell, Mac	organic anions
* Slc4a7 *	B cell	sodium-bicarbonate symporter
* Slc39a1 *	B cell	Zn^2+^
* Slc13a3 *	EnC, B cell, Mac	succinate and other Krebs cycle intermediates
* Slc26a2 *	B cell	bicarbonate, chloride, oxalate, sulfate
* Slc12a7 *	B cell	ammonium, potassium-chloride symporter
* Slc8a1 *	CD-PC	Ca^2+^
* Slc25a30 *	CD-PC	C4-dicarboxylate, sulfur compound
* Slc2a4 *	CD-PC	glucose, insulin-responsive glucose-proton symporter
* Slc6a6 *	CD-PC	beta-alanine, taurine
* Slc2a1 *	CD-PC	glucose, dehydroascorbic acid, long-chain fatty acid
* Slc44a1 *	CD-PC, EnC	choline
* Slc35a1 *	EnC	CMP-N-acetylneuraminate, pyrimidine nucleotide-sugar
* Slc39a3 *	EnC	Zn^2+^
* Slc35a4 *	Mac	pyrimidine nucleotide-sugar
* Slc7a8 *	Mac	glycine
* Slc35f5 *	Mac	NA
* Slc16a1 *	Mac	Lactate, carboxylic acid, succinate
* Slc22a5 *	Mac	carnitine, nucleotide
* Slc9a3 *	Mac	Na^+^-H^+^, K^+^-H^+^
* Slc20a2 *	Mac	inorganic phosphate
* Slc1a1 *	Mac	aspartate, glutamate, chloride, cysteine

NA—not available.

**Table 5 ijms-24-13502-t005:** ESDEG involved in corresponding cell communication.

Cluster	Role in Communication	Pathway	DEG
DLH_HDC	Source	WNT, COLLAGEN, AGRN, APP, L1CAM	*Wnt7b*, *Col4a3*, *Col4a4*, *Col4a5*, *Agrn*, *App*, *L1cam*
DLH_DDC	Source	EGF, COLLAGEN, AGRN, APP	*Egf*, *Fgf1*, *Col4a3*, *Col4a4*, *Col4a5*, *Agrn*, *App*
CD-PC_HDC	Source	KIT, COLLAGEN, APP, JAM, F11r	*Kitl*, *Col4a4*, *App*
CD-PC_DDC	Source	KIT, COLLAGEN, APP, JAM	*Kitl*, *Col4a4*, *App*, *F11r*
EnC_HDC	Source	VEGF, VISFATIN, COLLAGEN	*Vegfa*, *Nampt*, *Col4a4*
EnC_DDC	Source	TGFb	*Tgfb1*
B cell_HDC	Source	MHC-II	*H2-Ab1*, *H2-Ob*
B cell_DDC	Source	TGFb, MHC-II, SELL	*Tgfb1*, *H2-Ab1*, *H2-Ob*, *Sell*
Mac_HDC	Source	MIF, SPP1, COLLAGEN, JAM	*Mif*, *Spp1*, *Col4a3*, *F11r*
Mac_DDC	Source	MIF	*Mif*
DLH_HDC	Target	ANGPTL, COLLAGEN, FN1, AGRN, LAMININ	*Sdc4*, *Dag1*
DLH_DDC	Target	ANGPTL, COLLAGEN, FN1, AGRN, LAMININ	*Sdc4*, *Dag1*
CD-PC_HDC	Target	ncWNT, JAM	*Fzd7*, *F11r*
CD-PC_DDC	Target	JAM	*F11r*
EnC_HDC	Target	NULL	NULL
EnC_DDC	Target	NULL	NULL
B cell_HDC	Target	APP	*Cd74*
B cell_DDC	Target	APP	*Cd74*
Mac_HDC	Target	JAM	*F11r*
Mac_DDC	Target	NULL	NULL

## Data Availability

Renal scRNA-seq data of 4 Ctrl mice were adopted from NCBI GEO database (GSE107585) and we had registered scRNA-Seq data of 4 DKD kidneys in Sequence Read Archive (SRA, access number: PRJNA749372). All other intermediate documents, data and materials are available in the corresponding sections and Appendix A of current article without any restraint. The bulk RNA-Seq data had been uploaded in SRA (PRJNA945213).

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
