# Peer review of "Single-Cell Transcriptional Landscape Reveals the Regulatory Network and Its Heterogeneity of Renal Mitochondrial Damages in Diabetic Kidney Disease"

_ijms, 2023, doi:10.3390/ijms241713502_

Round 1
Reviewer 1 Report
The authors have assembled a detailed manuscript and provided a valuable data resource for future research in mouse DKD. The exploration of mitochondrial biology is an often under valued process in single cell sequencing. This work is deserving of publication in IJMS in it's current form to provide a single cell data resource and up to date analysis strategy for the readership of this journal.
Author Response
Dear Editor and Reviewers,
We wish to thank the reviewers for their positive consideration and critical comments!
According to the comments, we have revised this manuscript mainly including:
1 We have shorted down the label abbreviation and adjusted the order.
2 We accepted the reviewer's suggestion and adjusted the layout of some pictures in figures readable.
3 We have checked out the grammar in English.
We hope you find the revised manuscript suitable for publication in your journal IJMS!
Best regards
Sincerely,
Harvest F Gu
Reviewer 2 Report
The authors leveraged in particular scRNA-seq datasets for diseased and healthy control mice for diabetic kidney disease (DKD). Interestingly, they identify variable mitochondrial stress patterns across inferred cell types in disease mice, indicative of different disease statuses across the cell population. They show that comparing cells inferred as 'healthy' vs those inferred as 'disease' in diseased mice often give different insights than when comparing 'healthy' cells from healthy mice to the 'disease' cells from the diseased mice. They then identify the genes exclusive to the first comparison and perform more in-depth analyses. They identify DKD-associated communication pathways enriched in the "diseased" portion of the cells as well s TF networks and KEGG pathways potentially disrupted in DKD.
Overall the analysis is thorough, the methods well explained and involve computational approaches which are generally well established in the single-cell transcriptomics community. The results and discussion appear detailed and insightful - I am however not an expert in diabetic kidney disease, so do not consider myself qualified to judge these insights.
Furthermore, there is a heavy use of abbreviations in this article - causing me to have to backtrack and re-read parts of the text many times. Section 3.4 was particularly challenging (but it is an issue throughout). I ended up writing a table to help me keep track of them all and it helped. Perhaps they are not all necessarily, but a Table or list somewhere in the text might be useful for the reader. Also, please make sure they are defined at first use in the Methods section.
Finally - some of the text in the figures is hard to read - e.g. Figure 5 A/E, 4C and 6A (Figure 2 is much better). This could be due to the density of information in each figure - one strategy might be to move some plots to the supplementary to keep more room to others.
The overall English standard is good but as a native speaker I detect quite a few grammatical errors, which often break the flow of the text. It would be worth having a native speaker to go over it.
Author Response
Dear Editor and Reviewers,
We wish to thank the reviewers for their positive consideration and critical comments!
According to the comments, we have revised this manuscript mainly including:
1 We have shorted down the label abbreviation and adjusted the order.
2 We accepted the reviewer's suggestion and adjusted the layout of some figures readable particularly in Figures 4, 5 and 6.
3 We have checked out the grammar in English.
We hope you find the revised manuscript suitable for publication in your journal IJMS!
Best regards
Sincerely,
Harvest F Gu